# Evaluation of the Role of Leisure Time Physical Activity and Sedentary Behavior Simultaneously on the Income-Overweight/Obesity Relationship

**DOI:** 10.3390/ijerph18063127

**Published:** 2021-03-18

**Authors:** Layton Reesor-Oyer, Rosenda Murillo, Emily C. LaVoy, Daniel P. O’Connor, Yu Liu, Daphne C. Hernandez

**Affiliations:** 1Department of Exercise Science, Arnold School of Public Health, University of South Carolina, Columbia, SC 29208, USA; 2Department of Psychological, Health, and Learning Sciences, University of Houston, Houston, TX 77004, USA; rmurillo3@uh.edu (R.M.); yliu207@central.uh.edu (Y.L.); 3Health Research Institute, University of Houston, Houston, TX 77004, USA; dpoconno@central.uh.edu; 4Department of Health and Human Performance, University of Houston, Houston, TX 77004, USA; eclavoy@uh.edu; 5Department of Research, Jane and Robert Cizik School of Nursing, The University of Texas Health Science Center, Houston, TX 77030, USA; daphne.hernandez@uth.tmc.edu

**Keywords:** body mass index, federal poverty level, National Health and Nutrition Examination Survey, elevated weight status, exercise, health disparities

## Abstract

In the United States, overweight/obesity is more prevalent among those with low-income; higher income is related to greater leisure time physical activity (LTPA) and sedentary behavior (SB), which are inversely related to overweight/obesity. This study aimed to evaluate the role of LTPA and SB simultaneously in the income-overweight/obesity relationship. Cross-sectional data from the National Health and Nutrition Examination Survey (2007–2014) were utilized (*n* = 10,348 non-older adults (aged 20–59 years)). A multiple mediator structural equation model was conducted to evaluate the indirect effects from income to overweight/obesity (Body Mass Index ≥25 kg/m^2^) through LTPA and SB simultaneously, controlling for confounding variables, including diet, smoking, and alcohol consumption. As expected, greater income was negatively associated with overweight/obesity. Income indirectly influenced overweight/obesity through LTPA (Indirect effect: B = −0.005; CI = −0.01, −0.003), and through SB (Indirect effect: B = 0.008; CI = 0.005, 0.01), in opposing directions. The direct effect from income to overweight/obesity remained statistically significant. LTPA partially accounted for the negative relationship between income and overweight/obesity; SB reduced the strength of the negative relationship between income and overweight/obesity. Targeted behavior approaches for weight management may be beneficial. Increasing LTPA among adults with lower income and decreasing SB among adults with higher income may provide some overweight/obesity protection.

## 1. Introduction

Seventy percent of U.S. adults are considered overweight or obese [1], with obesity being more prevalent among those with low-income [2]. For example, the prevalence of overweight and obesity is higher among low-income households (74%) [i.e., income 100–199% of the Federal Poverty Line (FPL)] compared to those whose household income is 400% FPL or greater (66%) [3]. The health consequences associated with overweight and obesity [4] make it important to prevent and reduce overweight/obesity. Obesity prevention programs are designed to target lifestyle behaviors that are modifiable, regardless of an individual’s income bracket. A better understanding of how modifiable lifestyle behaviors, such as leisure time physical activity (LTPA) and sedentary behavior (SB) are related to the income-overweight/obesity relationship will inform obesity prevention programs tailored to those with low-income.

LTPA is an important behavior for obesity prevention [5]. LTPA is known to positively impact health; adequate levels are associated with decreased risk of obesity and chronic disease U.S. Department of Health and Human Services, [6]. LTPA is protective against obesity; those who engage in greater LTPA have decreased risk of overweight/obesity, even when controlling for energy intake [7]. In general, there appears to be a linear relationship with body mass index (BMI) and LTPA, such that those with higher BMIs engage in less LTPA [8]. Despite these benefits, many U.S. adults are insufficiently active [9]. Further, disparities exist with those of low-income engaging in less LTPA [10,11]. Only 41% of adults with an income below USD 15,000 met the PA guidelines from LTPA compared to 59% of adults with an income of USD 75,000 or greater [9]. 

Another modifiable lifestyle behavior associated with overweight/obese weight status is SB. In contrast to LTPA, SB is positively associated with overweight/obesity [12,13,14,15]. SB is known to have health consequences (all-cause mortality, cardiovascular disease, cancer, type 2 diabetes incidence) independent of physical activity [16]. Specific guidelines for SB do not exist, but individuals are encouraged to minimize SB [17]. The relationship between income and SB is less clear than the relationship of income and LTPA. Overall, it appears that greater income is related to greater total time spent in SB. For example, Kozo et al. [18] found that residents of higher income neighborhoods spent more objectively measured time in SB than those living in lower income neighborhoods. However, the relationship between income and SB differs when evaluating specific types of SB (e.g., television watching vs. occupational sitting) [15]. Some studies suggest that leisure time SB is more strongly related to poor health outcomes than occupational SB [19].

Although related, LTPA and SB represent two distinct concepts. It is possible for individuals to be highly active (e.g., meet/exceed PA recommendations), and yet spend many hours per day in SB, such as in a desk job. The inverse is also possible. Individuals may spend many hours per day in light intensity activity, but not necessarily moderate to vigorous intensity physical activity necessary to meet the PA guidelines, yet spend very little time engaging in SB. Overall the research indicates that time spent in SB is inversely related to physical activity and the behaviors differ based on household income [15]. While research has focused on these behaviors concurrently in relation to weight status, there is a lack of research evaluating LTPA and SB simultaneously in relation to the income-overweight/obesity relationship. As LTPA and SB are known to be inversely related to each other [15], and expected to differentially impact the income-overweight/obesity relationship it is important to understand their roles in the income-overweight/obesity relationship simultaneously.

Given that income-related overweight/obesity disparities exist in the U.S., and LTPA and SB are related to both income and weight status, it is possible that these behaviors play an important role in the income-overweight/obesity relationship. However, there are a lack of studies evaluating the role of LTPA and SB simultaneously in the income-overweight/obesity relationship. Multiple mediator structural equation modelling is needed in order to address this gap in the literature. The purpose of this study was to understand the modifiable lifestyle behavior mechanisms by which income influences overweight/obesity. Specifically, this study utilized a multiple mediator structural equation model to evaluate the role of LTPA and SB simultaneously controlling for the influence of one another in the income-overweight/obesity relationship utilizing a U.S. nationally representative sample with directly assessed measures of weight status. The first aim was to evaluate the indirect effect of LTPA on the income-overweight/obesity relationship, controlling for SB. Building on prior literature, it was expected that higher income would be positively related to LTPA [10,11] and LTPA would be negatively related to overweight/obesity [5,7,8]. It was further hypothesized that there would be a negative indirect effect from income to overweight/obesity through LTPA, which would partially account for the overall negative association between income and overweight/obesity. The second aim was to evaluate the indirect effect of SB on the income-overweight/obesity relationship, controlling for LTPA. Additionally, building on prior literature it was expected that higher income would be positively related to SB [18] and SB would be positively related to overweight/obesity [12,13,14,15]. It was further hypothesized that there would be a positive indirect effect from income to overweight/obesity through SB, working in the opposite direction of LTPA and the overall negative association between income and overweight/obesity. Although it was expected that there would be significant indirect effects through LTPA and SB, it was also expected that there would still be a significant direct effect from income to overweight/obesity [2] because of the complex multifaceted nature of this relationship. 

## 2. Methods

### 2.1. Dataset and Sample

This study utilized publicly available deidentified data from The National Health and Nutrition Examination Survey (NHANES) [20]. The NHANES is a cross-sectional study, which combines surveys, examinations, and lab measures to assess health and nutrition in the United States population. NHANES uses a complex, multistage stratified probability cluster sample design to obtain a nationally representative sample of the non-institutionalized U.S. civilian population [21]. The present study includes participants from four NHANES waves (2007–2014). These four waves of data contain consistent measures of physical activity variables and yielded a total of 40,617 adults and children. The initial sample was reduced to a non-pregnant adult sample (ages 20 and over) (*n* = 23,235). Further, the analytical sample was reduced to only include adults aged 20–59 (*n* = 15,376). Age 59 was selected as the cut-off because the Administration on Aging refers to individuals over the age of 60 as older adults, who may have behavioral and physiological differences from their younger counterparts [22]. Those with PA values deemed unrealistic (values > 3 standard deviations above the mean; approximately 25 h per week) were eliminated from analyses (*n* = 226). Additionally, due to small sample size those with a BMI value categorized as underweight (BMI < 18.5 kg/m^2^, underweight category) were excluded from the analytical sample (*n* = 261). Finally, only those with complete data on all target variables were included in the analytical sample. Individuals with missing data on the following variables were excluded: body mass index (*n* = 609), income (*n* = 1193), LTPA (*n* = 14), SB (*n* = 27), and control variables (*n* = 2698). The final analytical sample consisted of 10,348 adults. Informed consent was obtained by the NHANES team; this study was approved by the Institutional Review Board (or Ethics Committee) of The University of Houston (STUDY00000815 03/19/2018).

### 2.2. Overweight/Obese Weight Status

Height and weight were directly assessed by NHANES. Body Mass Index (BMI) was calculated as kg/m^2^. Individuals were classified as underweight (BMI < 18.5 kg/m^2^), normal weight (18.5–24.9 kg/m^2^), overweight (25–29.9 kg/m^2^), and obese (BMI > 30 kg/m^2^) based on CDC guidelines [23]. The overweight and obese categories were collapsed in order to compare those with an elevated weight status to those with a normal weight status [overweight/obese vs. normal (reference)].

### 2.3. Income

Income is a continuous measure based on the Federal Poverty Level (FPL), such that higher values correspond to higher incomes. The Department of Health and Human Services issues the FPL based on annual average estimates of the cost to cover basic needs. Income level for each participant was calculated by NHANES dividing self-reported annual household income by the FPL corresponding to the number of individuals residing in the household. An income level of less than 1 is considered to be poor.

### 2.4. Leisure Time Physical Activity

Participants self-reported the amount of time they typically engage in moderate or vigorous intensity activity from “sports, fitness and recreational activities”. The variable was coded as weekly hours (continuous) and an equivalent combination of moderate and vigorous-intensity LTPA was calculated by assigning vigorous intensity activities twice the weight of moderate-intensity activity as suggested by the 2018 Physical Activity Guidelines for Americans [24]. Higher values correspond to greater time spent in moderate to vigorous LTPA. 

### 2.5. Sedentary Behavior Time

Participants self-reported the amount of time they typically spend sitting or reclining excluding sleep per day. The variable was coded as daily hours (continuous), such that higher values indicate more hours spent in sedentary activities. 

### 2.6. Control Variables

The following socio-demographic variables, known to be related to income, LTPA, SB, and overweight/obesity were self-reported through a survey: age (years), sex [male vs. female (reference)] race/ethnicity [black, white (reference), Hispanic, other] nativity status [foreign vs. native (reference)], marital status [single vs. married/cohabitating (reference)], education [less than high school diploma, high school diploma, college degree or greater (reference)], employment [unemployed vs. employed (reference)], health insurance coverage [does not have health insurance coverage vs. has health insurance coverage (reference)]. Several health behaviors were also included as covariates: Healthy Eating Index [25] (HEI; continuous, higher values indicate a healthier diet), alcohol consumption (drinks per day continuous, higher values indicate a greater number of average alcoholic beverages per day), smoking status [smoker vs. non-smoker (reference)], and sleep (average hours per night; continuous, higher values indicate a greater number of average hours the participant sleeps per night).

### 2.7. Statistical Analyses

Means, frequencies and standard errors of participant characteristics were computed for the full sample and by BMI category (overweight/obese vs. normal). Adjusted Wald Tests were used to determine differences by BMI category. Descriptive statistics and adjusted Wald Tests were conducted using Stata SE version 15.0 statistical software (StataCorp, College Station, TX, USA). To test whether LTPA and SB contributed uniquely to the relationship of income and overweight/obesity in combination with each other, a multiple mediator structural equation model was conducted. Standardized estimates are presented; 95% bootstrapped confidence intervals (5000 resamples) were utilized to determine statistical significance of the indirect paths. Standardized beta coefficients are presented for continuous outcomes and indirect effects; odds ratios are presented for dichotomous outcomes. The residual errors of LTPA and SB were correlated. Structural equation models were conducted in Mplus version 8.3 (Muthen & Muthen, Los Angeles, CA, USA). Survey procedures were used to account for the complex NHANES sampling design.

## 3. Results

### 3.1. Descriptive Statistics

Characteristics of the full sample and by weight status are presented in Table 1. Sixty nine percent of participants were classified as overweight/obese. The average income (FPL) of the sample was 3.04 (0.05). Participants reported engaging in 3.48 (0.10) hours per week of moderate to vigorous LTPA and 6.25 (0.07) hours per day of SB. The average age of the sample was 39.47 (0.23) years. Sixty-seven percent of the sample was white, followed by Hispanic (15%), black (11%), and another race/ethnicity (7%). Seventeen percent of the sample was foreign born, 63% were married or cohabitating, and 55% had a high school diploma but not a college degree. Twenty-three percent of the sample did not have health insurance. Participants reported an average HEI score of 52.54 (0.27) and consumed an average of 0.60 (0.02) alcoholic beverages per day. Twenty-three percent of the sample was categorized as a smoker. 

Differences were detected by weight status. Participants classified as overweight/obese had lower income (*p* < 0.05), engaged in less LTPA (*p* < 0.001), and more time in SB (*p* < 0.001) than their counterparts classified as normal weight. Those classified as overweight/obese were older (*p* < 0.001); a greater percentage were male (*p* < 0.001), black (*p* < 0.001), and Hispanic (*p* < 0.001), but a lower percentage were white (*p* < 0.01), or another race (*p* < 0.001), compared to those classified as normal weight. A greater percentage of those classified as overweight/obese were married/cohabitating (*p* < 0.001), had less than a high school education (*p* < 0.01), or a high school education (*p* < 0.001), but a lower percentage had a college degree or greater ((*p* < 0.001). HEI scores (*p* < 0.001),) and alcoholic beverage consumption (*p* < 0.05) were lower among those classified as overweight/obese than those classified as normal weight. Smoking was less common among those classified as overweight/obese than those classified as normal weight (*p* < 0.001).

### 3.2. Structural Equation Modeling

Standardized estimates are presented; 95% bootstrapped confidence intervals (5000 resamples) are included for the total, and indirect effects. Greater income was associated with decreased risk of overweight/obesity (Total effect: B = −0.043; CI = −0.07, −0.02). In the context of the overall model, income was positively associated with LTPA (B = 0.06, SE = 0.01; *p* < 0.001), and greater LTPA was associated with decreased risk of overweight/obesity (OR = 0.97, SE = 0.01; *p* < 0.001). The indirect effect from income to overweight/obesity through LTPA was statistically significant (specific indirect: B = −0.005; CI = −0.01, −0.003). LTPA partially accounted for the negative relationship between income and overweight/obesity.

Further, in the context of the overall model income was positively associated with SB (B = 0.08, SE = 0.01, *p* < 0.001), and greater SB was associated with increased risk of overweight/obesity (OR = 1.05, SE = 0.01, *p* < 0.001). The indirect effect from income to overweight/obesity through SB was also statistically significant (specific indirect: B = 0.008; CI = 0.005, 0.01). SB reduced the negative relationship between income and overweight/obesity. The direct effect from income to overweight/obesity remained statistically significant (OR = 0.91, SE = 0.02, *p* < 0.001) such that greater income was associated with decreased risk of overweight/obesity. See Figure 1. The specified model contains the maximum number of possible pathways (“just identified” model); for this reason, we were unable to assess model fit indices. Indirect effects are presented in Table 2.

## 4. Discussion

The purpose of this study was to understand the modifiable lifestyle behavior mechanisms by which income influences overweight/obesity. Specifically, this study evaluated the role of LTPA and SB simultaneously, controlling for one another in the income-overweight/obesity relationship. The hypotheses were supported. As expected, higher income was positively related to LTPA [10,11] and LTPA was negatively related to overweight/obesity [5,7,8]. The significant indirect effect from income to overweight/obesity through LTPA indicated that greater LTPA among those with higher income may partially explain how having a higher income is protective against overweight/obesity. Further, as expected income was positively related to SB [18] and SB was positively related to overweight/obesity [12,13,14,15]. Greater SB among those with higher income buffered the negative association between income and overweight/obesity. The indirect effect of SB worked in the opposite direction of LTPA and the overall negative association between income and overweight/obesity. Additionally, in line with the hypotheses, there was a significant negative direct effect from income to overweight/obesity, such that those with greater income were at decreased odds having an overweight/obese weight status [2]. This indicates that the relationship between income and overweight/obesity is not entirely accounted for by LTPA and SB. In fact, the direct effect from income to overweight/obesity was much larger than the indirect effects through LTPA and SB. This is unsurprising because the income-overweight/obesity relationship is complex and influenced by several factors. 

This study sheds light on two modifiable health behaviors related to the income-overweight/obesity relationship. It informs healthcare practitioners attempting to address overweight/obesity among both high and low-income populations. According to the results, there was a significant negative indirect effect from income to overweight/obesity through LTPA. This indicates that LTPA may partially account for the negative relationship of income and overweight/obesity. Thus, higher participation in LTPA may partially explain the lower prevalence of overweight/obesity among those with higher income, while lower participation in LTPA may partially explain the greater prevalence of overweight/obesity among those with lower income. How this mechanism may function is as follows. Those with higher income may utilize their resources to engage in LTPA [26]. For example, individuals with high incomes may use their funds to purchase memberships at fitness facilities, to live in areas with greater walkability, or increased access to places, such as parks and trails that promote LTPA. Some individuals with higher income may use their funds to outsource household chores (cleaning, lawn care, etc.) or live closer to their place of employment (e.g., shorter commute); thus, having more available time to engage in LTPA. Whereas those with lower income may not have the discretionary funds to invest in facilities nor the time for LTPA. Additionally, greater education among those with higher income may be a factor. In this sample, those with a college degree or greater had a much higher income (FPL M = 4.02, SE = 0.05) than those with less than a high school degree (FPL M = 1.76, SE = 0.06; *p* < 0.001), or a high school degree (FPL M = 2.82, SE = 0.05; *p* < 0.001). Previous studies have found that those with higher educational attainment engage in greater LTPA [9]. Those with greater educational attainment may be able to more easily understand and interpret health literature and therefore engage in more health promoting behaviors, such as LTPA. 

In contrast, there was a significant positive indirect effect from income to overweight/obesity through SB. The indirect effect through SB is in the opposite direction of the overall effect of income on overweight/obesity. SB reduces the strength of the negative relationship of income and overweight/obesity. However, it does not reduce the negative association completely. In fact, a lower amount of time spent in SB among those with lower income appears to be protective against having an overweight/obese weight status. The relationship between income and SB is likely a reflection of different types of occupations. “White collar” or “professional” occupations associated with higher socioeconomic status are considered less active and higher in SB compared to “blue collar” occupations, traditionally considered lower socioeconomic status [27,28,29,30,31]. Further, some individuals with lower income may engage in more household or transportation physical activity due to a lack of resources and therefore spend less time in SB.

Although previous studies have not evaluated the indirect effect of income on overweight/obesity through LTPA and SB, this study aligns with others who evaluated specific pathways of interest. The findings that income was associated with decreased risk of overweight/obesity [2], positively associated with LTPA [10,11] and that LTPA was associated with decreased risk of overweight/obesity [5,7,8] are not novel. Nor are the findings that income was positively related to SB [18] and SB was associated with increased risk of overweight/obesity [12,13,14,15]. However, the finding that income indirectly affects overweight/obesity through *both* LTPA and SB is novel and important. A better understanding of modifiable health behaviors related to the income-overweight/obesity relationship will help healthcare practitioners develop targeted approaches for weight management.

### Limitations

This study attempted to understand the mechanisms by which income influences overweight/obesity, specifically LTPA and SB. In doing so, this study included a number of covariates in order to isolate the indirect effects from income to overweight/obesity through LTPA and SB, including a number of modifiable lifestyle behaviors such as diet (HEI), alcohol consumption, smoking status, and sleep. However, the relationship between income and overweight/obesity is complex, and there are numerous factors, which may be involved in this relationship that were not available in the dataset. This includes measures of the neighborhood environment, such as walkability [32] and proximity to fast food [33]. Additionally, the cross-sectional design of the publicly available data prevents one from establishing the directionality of the study variables. However, this study does lay the groundwork for future studies to examine the association between income, LTPA, SB, and overweight/obesity using longitudinal data and randomized controlled trials. Further, there are known limitations with self-reported LTPA and SB data, specifically over-reporting of LTPA [34]. Future studies are encouraged to utilize objective measures when evaluating LTPA and SB. Despite these known limitations, there are also strengths of utilizing NHANES. For example, the NHANES study sample is designed to be nationally representative, making the findings highly generalizable. Further, weight status was directly assessed by trained research staff, rather than self-reported, which is known to have error.

## 5. Conclusions

This study provides a framework for understanding the role of LTPA and SB in the income–overweight obesity relationship. Findings from this study indicate that greater LTPA among those with higher income partially accounts for the lower prevalence of overweight/obesity among those with higher income. Restated, lower LTPA among those with lower income partially accounts for the higher prevalence of overweight/obesity among those with lower income. In contrast, SB reduces the strength of the income-overweight/obesity relationship. SB works in the opposite direction of the overall negative association of income with overweight/obesity, buffering the influence of income on overweight/obesity. A lower amount of time spent in SB among those with lower income provides some protection against overweight/obesity among this vulnerable population. Thus, healthcare practitioners attempting to address overweight/obesity among those with higher income may want to consider focusing on decreasing SB, rather than increasing LTPA, which appears to be protective against overweight/obesity. In contrast, healthcare practitioners attempting to reduce overweight/obesity among those with lower income may want to consider focusing on increasing LTPA, rather than decreasing SB. Providing education in isolation is unlikely to address LTPA disparities among low-income populations. In order to support individuals with lower income in increasing LTPA and reducing overweight/obesity both individual and structural level interventions may be necessary to overcome barriers to engagement in LTPA. For example, those with lower income may not have the discretionary funds to invest in facilities for LTPA. Thus, ensuring that safe and affordable locations to engage in LTPA are accessible to individuals with lower income is necessary. Randomized controlled trials to decrease overweight/obesity among income-specific populations using targeted approaches to overcome barriers to increase LTPA among those with low-income and decreasing SB among those with higher-income are needed to better understand the efficacy of targeting specific health behaviors by income status. The results from this study lay the groundwork for future research studies utilizing more sophisticated approaches (e.g., randomized controlled trials with objective measures of activity) to understand the roles of LTPA and SB in the income–overweight/obesity relationship.

## Figures and Tables

**Figure 1 ijerph-18-03127-f001:**
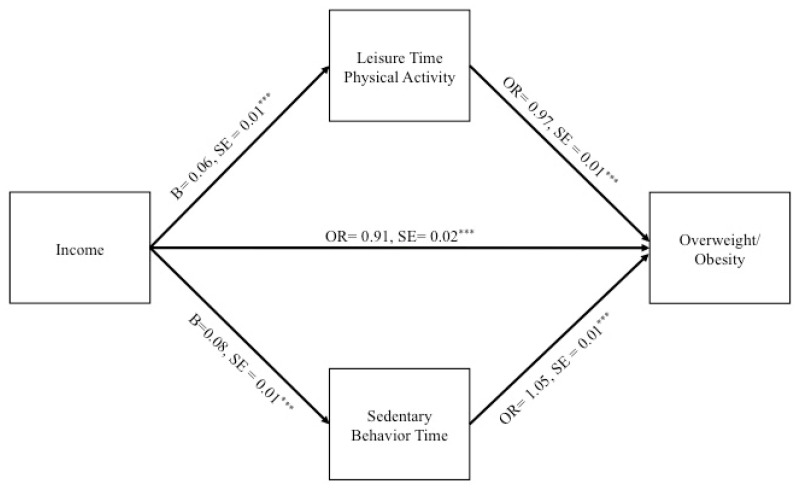
Multiple Mediator Structural Equation Model Assessing the Role of Leisure Time Physical Activity and Time in Sedentary Behaviors on the Income-Overweight/Obesity Relationship. Total effect: B = −0.043; CI = −0.07, −0.02; *** *p* < 0.001; NOTE: The specified model contains the maximum number of possible pathways (“just identified” model); for this reason, we were unable to assess model fit indices. Each pathway includes the following covariates (not pictured): age, sex, race/ethnicity, nativity status, marital status, education, employment status, health insurance, Healthy Eating Index, alcoholic beverage consumption, and smoking status. The residual errors of leisure time physical activity and sedentary behavior time were correlated (not pictured).

**Table 1 ijerph-18-03127-t001:** Characteristics of participants by weight status: National Health and Nutrition Examination Survey 2007–2014, M (SE) or %.

	Full Sample (*n* = 10,348)	Normal Weight (*n* = 3136)	Overweight/Obese (*n* = 7212)	F	Effect Size Cohen’s D or Odds Ratio
Dependent variable					
Weight status ^					
Overweight/obese	69%	---	---		
Normal weight	31%	---	---		
Independent variable					
Income (FPL) ^	3.04 (0.05)	3.11 (0.07)	3.01 (0.05) *	4.08	0.02
Mediating variables					
Leisure time physical activity (weekly hours)	3.48 (0.10)	4.30 (0.17)	3.11 (0.09) ***	58.37	0.14
Sedentary behavior (Daily hours)	6.25 (0.07)	5.98 (0.10)	6.37 (0.08) ***	12.41	0.06
Demographic characteristics					
Age	39.47 (0.23)	36.79 (0.41)	40.69 (0.22) ***	89.92	0.20
Sex					
Female	50%	56%	48% ***	42.78	0.72
Male	50%	44%	52%		
Race/ethnicity					
White	67%	70%	65% **	9.26	0.82
Black	11%	8%	13% ***	42.97	1.59
Hispanic	15%	11%	17% ***	33.11	1.64
Other	7%	11%	5% ***	43.68	0.44
Nativity status					
Foreign born	17%	19%	17%	3.44	1.17
Native born	83%	81%	83%		
Marital status					
Single	37%	42%	35% ***	24.59	0.74
Married/cohabiting	63%	58%	65%		
Education					
Less than high school degree	14%	12%	15% **	11.67	1.29
High school degree	55%	50%	56% ***	15.05	1.26
College graduate or greater	31%	37%	29% ***	30.93	0.67
Employment					
Employed	74%	73%	75%	2.54	1.09
Unemployed	26%	27%	25%		
Health insurance					
Insured	77%	77%	75%	0.02	1.01
Uninsured	23%	23%	25%		
Health behaviors					
Healthy Eating Index	52.54 (0.27)	54.08 (0.43)	51.84 (0.25) ***	35.31	0.10
Average alcoholic drinks per day	0.60 (0.02)	0.66 (0.04)	0.58 (0.02) *	4.52	0.04
Smoking status					
Smoker	23%	27%	21% ***	15.37	0.74
Non-smoker	77%	73%	79%		
Average hours sleep per night	6.83 (0.02)	6.97 (0.03)	6.76 (0.02) ***	39.63	0.12

* *p* < 0.05, ** *p* < 0.01, *** *p* < 0.001; ^ Those with a body mass index 18.5 to 24.9 were classified as normal weight; those with a body mass index > 24.9 were classified as overweight/obese; FPL federal poverty level.

**Table 2 ijerph-18-03127-t002:** Indirect effect of leisure time physical activity and sedentary behavior time on the association between income and overweight/obesity (*n* = 10,348).

Indirect Effect	B	SE	95% Bootstrap CI
Income → Leisure time physical activity → Overweight/obesity	−0.005	0.001	−0.01, −0.003
Income → Sedentary behavior time→ Overweight/obesity	0.008	0.002	0.005, 0.01

Note: B = Standardized Beta Coefficient, SE = Standard Error, CI = Confidence Interval.

## Data Availability

The data supporting reported results are publicly available and can be found at https://www.cdc.gov/nchs/nhanes/index.htm (accessed on 1 December 2018).

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
