# Peer review of "Evaluation of the Role of Leisure Time Physical Activity and Sedentary Behavior Simultaneously on the Income-Overweight/Obesity Relationship"

_ijerph, 2021, doi:10.3390/ijerph18063127_

Round 1

Reviewer 1 Report

(1) In the introduction, authors need to describe the research problem, the gap of knowledge in the preceding studies to be filled, and the excellence of this paper in sufficient detail.

(2) The explanation of the research model and hypothesis shown in Figure 1 is largely insufficient. In other words, the validity and logical basis of the research model based on previous studies should be sufficiently supplemented.

(3) A lot of detailed results such as analysis figures for statistical analysis results were omitted.

(4) The meaning and interpretation of measurement variables should be reinforced.

(5) In the conclusion, a summary of theoretical and practical contributions, implications and insights, and directions for future research should be sufficiently additionally described.

(6) The resolution in Table 1 is very poor. Authors are encouraged to create and include Table 1 directly in the manuscript.

(7) In the abstract, too detailed numerical results and abbreviations were used. Authors need to revise the abstract more concisely.

Reviewer 2 Report

First, congrats for this manuscript. The paper is well written, the introduction prepares the reader for the methods and results. Finally, the discussion is well written with deep information supporting the results. I firstly present some minor commentaries:

L.128: Include BMI units.

LL.135-138: Include BMI units

L.196-207: Include the T values and effect sizes.

L.208. Table 1 is an image. The authors may create the table and do not paste an image.

(a) and (b) below line 248 are errors?

In the discussion section, please include one paragraph discussing and supporting the methods.

The paper is well written. However, literature allow to predict the outcome. Moreover, this theme is not novel, and the novelty is focused on the statistical analysis. Statistics are methods to solve problems and not to create them. That said, the results are expected and this reviwer did not understand how this paper helps to fulfil the research gap. Actually, I did not understand the research gap. Again, that can be explained by the possibility to predict the results of this study based on literature.

Reviewer 3 Report

The present manuscript entitled "The role of leisure time physical activity and sedentary behavior in the income-overweight / obesity relationship" is an interesting article, which has certain limitations that make it suitable for publication in this journal. The limitations are listed below point by point:
1. The authors note in the abstract at line 16, that: "Overweight / obesity is more prevalent among those with low-income". This statement is too categorical for a summary. There are not enough references or studies to be that resounding.

2. Figure 1 does not provide much information to be in the introduction. It does not follow the magazine format for figures, as it does with Figure 2.

3. Line 112. Data are used from a national survey (NHANES). This survey is not cited or given information to access it. It is also not mentioned whether the data can be used, as it does not mention anything about the ethical committee.

4. Table 1 is a "copy and paste" from another document. Literally. Thus, a table cannot be presented for a scientific journal, and less with JCR.

Round 2

Reviewer 1 Report

(1) It is understood that the greatest contribution of this paper is considered to have simultaneously considered LTPA and SB as mediating parameters in the relationship between income and obesity, as the authors revealed. However, the authors only emphasize that LTPA and SB have never been considered at the same time in previous studies, and do not clearly explain why LTPA and SB should be considered. Why should LTPA and SB be considered as mediating parameters in the relationship between income and obesity compared to other variables? And in the relationship between income and obesity, what are the expected effects obtained by simultaneously considering LTPA and SB compared to each of the models? And is there a correlation between LTPA and SB? Obviously, the relationship between the variables is ambiguous, and it seems that it is not enough to clarify the meaning of the authors' hypothesis.

(2) Please add a paragraph at the end of Introduction to explain the whole structure of the manuscript.

(3) In the conclusion, further explanation is needed for the implications and insights of this study. In addition, a description of the research direction should be added later.

(4) The readability of the main results will increase when the statistical results are clearly organized in tables accordingly. 

Author Response

Thank you for your feedback

Reviewer 2 Report

The authors have adressed my concerns. The novelty of this study can be discussed and is the wekeast point of this manuscript. However, this is a wel-writen manuscript. 

Author Response

Thank you!

Reviewer 3 Report

The authors have been able to reflect the suggestions in the manuscript and have modified the manuscript correctly.

Author Response

Thank you!